# Voltage RMS Estimation during a Fraction of the AC Period

**DOI:** 10.3390/s22186892

**Published:** 2022-09-13

**Authors:** Ido Amiel, Zekharya Danin, Moshe Sitbon, Moshe Averbukh

**Affiliations:** Department of Electrical/Electronic Engineering, Ariel University, Ariel 40700, Israel

**Keywords:** AC voltage estimation, RMS value, harmonic representation, correcting coefficients

## Abstract

The increasingly widespread occurrences of fast-changing loads, as in, for example, the charging of electrical vehicles and the stochastic output of PV generating facilities, are causing imbalances between generated and consumed power flows. The deviations in voltage cause noteworthy technical problems. The tap-changers in today’s transformers are slow-reacting and thus cannot effectively correct the imbalance. Tap-changers should be replaced by special appliances, installed in distribution lines, that can effectively estimate voltage RMS and refine imbalances during a fraction of the AC period, preferably less than half. This article suggests specially developed methods for RMS assessment based on approximating instantaneous voltage magnitudes using harmonics and correcting coefficients.

## 1. Introduction

The problem of voltage correction in distribution lines has become more and more actual during the last ten years. The extensive dissemination of consumers having fast-changing loads, such as in-charge batteries in electric vehicles, electric trains being in the accelerating-decelerating modes, and so on, together with the stochastically generating plants (PV and wind power stations), has produced a significant challenge for distribution lines. This problem has been caused by significant stochastic energy flows and the inability of the transformer’s tap-changer to prevent dip and sag voltage deviations, which can cause inevitable problems for electrical/electronic equipment. Therefore, the grid should solve this problem by maintaining stable voltage levels. The scientific literature gives multiple examples of research works elaborating on the problem of voltage regulation and stabilization in distribution lines. Owing to the high voltage and power applications in distribution lines, the use of the reactive power concept seems like an attractive and efficient method [1,2,3,4,5,6,7,8,9,10,11]. Keddar et al. proposed reactive power sharing for voltage control [1], and Mehbodniya et al. [2] applied reactive power management in the smart distribution network enriched with wind turbines and photovoltaic systems. Iioka et al. [3] and Zimann et al. [4] suggested voltage improvement through current and reactive-active power injection into a grid. Ismail et al. provided a comprehensive review of reactive power compensation for voltage stability improvement and power loss reduction [5]. Aziz and Ketjoy used reactive power control for voltage stabilization during the enhancement of PV penetration in the grid [6]. Kim and Harley examined the effect of the reactive power control of photovoltaic systems on electric power grids for the development of an effective voltage-regulation method [7]. Ma et al. suggested with the same aim a reactive power optimization of substations [8], whereas Mirbagheri and Merlo proposed the optimization of a reactive power flow for voltage regulation in a grid [9]. Cárdenas et al. submitted a dynamic voltage restorer for voltage restoration using reactive power, and active filtering [10], whereas Hu et al. promote coordinated active and reactive power control for distribution networks [11]. Salah Saidi analyzed the actual voltage stability of Tunisian distribution networks with dynamic reactive power control [12].

Multiple research works are dedicated to the usage of capacitive reactive power for voltage control in distribution lines [13,14,15,16,17,18,19].

Feng and Fang [13], Mancer et al. [14], Nguyen et al. [15], and Pyo et al. suggested different electronic devices—condensers, compensators, or STATCOM appliances for the improvement of voltage control—whereas in [17,18,19] the authors submitted the usage of real capacitors for voltage regulation.

The improvement of voltage stability can be substantially increased with model predictive control (MPC) [20,21,22,23,24,25]. This type of control can provide the fastest reply to the changing conditions and imbalance between consumer power and generating abilities of a grid. It can be outdone, which is a pity to the interventive management of reactive power in a network for voltage amendment.

Valverde and Van Cutsem [20] suggested a control scheme to regulate network voltages in the presence of high penetration of distributed generation. The method is based on MPC to compensate for inaccuracies and measurement errors.

Kamal and Chowdhury [21] represented a review of the use of MPC in networks of distributed microgrids. The work was done due to the increasing requirements to develop more efficient voltage/frequency stability in microgrids with a sufficient penetration of renewable sources.

Babayomi et al. [22] presented the MPC of power electronic systems including DC/AC inverters of PV plants. The authors also analyzed MPC for energy management in autonomous and networked microgrids. The fast dynamic response and efficiency of the proposed control are especially emphasized.

Pan et al. [23] submitted an MPC scheme-based static synchronous compensator (STATCOMs) to regulate reactive power flow for compensating unbalanced voltage deviations. Hu et al. [24] and Dhulipala et al. [25] suggested distributed MPC to exploit reactive power for effective voltage regulation in distribution networks.

The MPC of reactive power will be especially effective if it is based on the rapid and accurate measurement of the voltage RMS value. The fast assessment of the RMS level will insure better correction of the voltage value. Fast RMS estimations of AC voltage are investigated in multiple works [26,27,28,29,30,31,32,33,34,35]. All suggested methods of RMS estimation can be represented by several large groups. The first one of these methods used a precise measurement and averaging of the rectified signal (Muciek [26]). Different kinds of frequency measurements for the voltage RMS value estimation are suggested in [27,28,29,30].

The hybrid method to quantify the magnitude of the voltage sag developed by Thakur and Singh [31] is based on an analysis of three existing methods, namely root mean square (RMS), peak, and fundamental voltage component assessment. A similar approach was suggested in [32] by Naidoo and Pillay.

Time-domain voltage sag estimation based on the unscented Kalman filter is described by Cisneros-Magaña et al. [33].

Flores-Arias et al. represented the RMS voltage estimator, which can be sufficiently applied in cases characterized by limited-resources hardware [34].

The new method for fast voltage detection under distorted grid voltage conditions was suggested in [35].

A summary of the analyzed methods is represented in Table 1.

The analysis of all submitted methods shows the lack of strict accuracy assessment for each approach in different estimation conditions. Since MPC reveals its advantages, especially in the prompt correction of voltage deviations (for example, during a fraction of an AC period), the need to determine the accuracy of a fast RMS estimation by each specific procedure is urgent.

This article gives a deep analysis of the main RMS voltage assessments including well-known standard RMS methods, as well as those developed in this work. The authors developed an original math method for the decomposition periodic signal by an arbitrary set of harmonic series. In addition, a combined approach of a fast RMS determination based on a restricted measuring time significantly less than half of the AC period was developed in this work. It was shown that for the effective correction, an RMS value should be estimated at a time that is less than the half of the AC period. The remaining time in the first half of each AC period can be necessary for the control system to calculate the required value of reactive power and to attach it to the grid before the beginning of the next half. This solution ensures the fastest correcting reaction for returning the voltage magnitude between the lowest and highest permissible levels.

The main contributions of work are the following:A deep analysis of the RMS magnitude assessment by the standard formulation and by peak voltage value was provided analytically and experimentally.Development of the original math approach for the decomposition of a function of an instantaneous voltage magnitude by harmonic series based on the solution of the system of linear algebraic equations.The development of a combined fast RMS determination approach that ensures the estimation during a restricted measuring time significantly less than half of the AC period and that is based on the application of a special coefficient obtained analytically as a function of the measuring time.The assessment of all represented methods for RMS estimation experimentally.

The article includes the following sections: (1) Introduction; (2) Description of methods of the fast RMS estimation, (3) Results, (4) Discussion, and (5) Conclusions.

Section 2 represents an analysis of the accuracy of the four main RMS assessment approaches; Section 2.1 presents a method of measuring by the voltage magnitude; Section 2.2 represents an instantaneous voltage measurement during the integer number of all AC periods; Section 2.3 presents the development of a method based on voltage magnitude measurement in only a fraction of the AC period, and Section 2.4 presents the development of a method based on fitting measured points with arbitrarily chosen voltage harmonics.

Section 3 represents the results of the experimental accuracy verification of all mentioned RMS estimation methods.

Section 4 summarizes the actuality of the work and provides the brief results.

Section 5 presents the conclusions of the article.

## 2. Methods of the Fast RMS Estimation

The four main methods to be analyzed are:
Measurement by voltage amplitude—maximum value in the AC period.Estimation due to strict math RMS determination through the entire AC period.Assessment through only a fraction of the AC period with a correcting coefficient.Fitting measured points with trigonometric functions representing voltage harmonics.

### 2.1. Measurement by Voltage Amplitude

Estimation of the RMS voltage value by means of this method (Figure 1) is carried out based on the instantaneous magnitude measured with the the expression describing RMS determination:(1)$$V_{\mathrm{RMS}}=\sqrt{\frac{1}{T}\int_{0}^{T}v^{2}\left(t\right)\mathrm{dt}}$$

Taking into consideration voltage as a sum n- sinus harmonics of a time, i.e.:(2)$$v\left(t\right)=\overset{n}{\underset{k=1}{\sum }}V_{\mathrm{mk}}\sin\left({k\omega }_{0}t\right)$$

Voltage RMS is calculated as:(3)$$V_{\mathrm{RMS}}=\sqrt{\frac{1}{2}\overset{n}{\underset{k=1}{\sum }}V_{\mathrm{mk}}^{2}}$$

As a rule, a voltage in distribution networks has a good harmonics quality and low THD value. Therefore, a good approximate RMS can be assessed with the amplitude of a main harmonic:(4)$$V_{\mathrm{RMS}}=\frac{U_{m}}{\sqrt{2}}$$

### 2.2. Estimation by Voltage Points Measured during the Integer Number of Entire AC Periods

Assume N equidistantly dispersed points of a voltage magnitude obtained during the integer number of the AC periods. This way the RMS value will be calculated as per its rigorous mathematical determination:(5)$$V_{\mathrm{RMS}}=\sqrt{\frac{1}{N}\left(\overset{N}{\underset{i=1}{\sum }}U_{i}^{2}\right)}$$

However, the requirement for a voltage correction of less than half of a period dictates the need for the RMS estimation in a fraction of the AC period.

### 2.3. Assessment through Only a Fraction of the AC Period

If the measured points of instantaneous voltage values were obtained during a fraction of an AC period only, the application of expression (5) leads to a rough error and is not applicable. For the correct RMS estimation expression, (5) should be modified.

For the AC frequency 50 Hz, the time of voltage measurements should be not less than 5 ms to overcome sinus amplitude. Owing to the requirements of a voltage correction during the half of an AC period, the measurement time cannot be more than 6–7 ms maximum. The remaining 3–4 ms to the half (10 ms) of a period are needed for the control system to evaluate a situation and decide which reactive power (capacitive or inductive) to correct voltage. If voltage RMS lies between permissible levels (+/−10% of its rated magnitude), the control system does not introduce reactive power into a network.

Assuming the measurement time from 5 ms to 10 ms, the time will be represented by the angle (βπ), where π is half of the period and β is the dimensionless coefficient. The expression for the RMS estimation will be:(6)$$V_{\mathrm{RMS}}^{*}=\sqrt{\frac{1}{\omega \theta }\overset{\theta }{\underset{0}{\int }}V_{m}^{2}\sin^{2}\left(\omega t\right)d\left(\omega t\right)}$$

After simplification and substitution (βπ) instead of variable θ:(7)$$V_{\mathrm{RMS}}^{*}=\frac{V_{m}}{\sqrt{2}}\sqrt{1-\frac{\sin\left(2\beta \pi \right)}{2\beta \pi }}$$

For voltage $U_{\mathrm{RMS}}^{*}$ to be identical to $U_{\mathrm{RMS}}=\frac{V_{m}}{\sqrt{2}}$ it must be multiplied by the correction coefficient K_C_ equal to:(8)$$\;\;{\;K}_{C}=\frac{1}{\sqrt{1-\frac{\sin\left(2\beta \pi \right)}{2\beta \pi }}}\;,{\;V}_{\mathrm{RMS}}=V_{\mathrm{RMS}}^{*}\cdot K_{C}$$

The dimensionless coefficient β value is the derivative of measuring time t_mes_:(9)$$\beta =\pi \frac{t_{\mathrm{mes}}}{10\;\mathrm{ms}},\;\;\;\mathrm{for}\;\mathrm{AC}\;\mathrm{frequency}\;50\mathrm{Hz}$$

The graph of a coefficient K_C_ vs. β value is represented in Figure 2.

When voltage measurement is represented by a set of M-instantaneous voltage points during a fraction of the AC period, for instance from a ¼ to ½ of its duration, the RMS assessment is made due to the following expression:(10)$$V_{\mathrm{RMS}}=K_{C}\cdot \sqrt{\frac{1}{N}\overset{N}{\underset{i=1}{\sum }}V_{i}^{2}}$$
where K_C_ is calculated due to (8) considering measuring time t_mes_ and β value from (9).

### 2.4. Fitting Measured Points with Trigonometric Functions Representing Voltage Harmonics

Another efficient method for the RMS estimation of periodic signals is the representation by a sum of k-first sinusoidal harmonics. The choice of sinusoid usage for the approximation is dictated by a form of a real voltage signal.

To estimate RMS with the proposed method, it is enough to obtain the harmonics amplitudes and furthermore to represent the RMS as:(11)$$V_{\mathrm{RMS}}=\frac{1}{\sqrt{2}}\sqrt{\overset{k}{\underset{i=1}{\sum }}{\left(V_{m}\right)}_{i}^{2}}$$

For the approximation of the measured points, a least-mean-square approach (LMS) will be applied:(12)$$S=\sum_{i=1}^{k}{ [V_{i}-A_{1}\sin\left({\omega t}_{i}\right)-A_{3}\sin\left(3{\omega t}_{i}\right)-\ldots -A_{2k-1}\sin\left(\left(2k-1\right){\omega t}_{i}\right)]}^{2}\to \mathrm{MIN}$$
where S—the criterion of approximation, V_i_—is the i-measured instantaneous voltages (from the set of N-voltage points) obtained in a t_i_ moment of a time, A_1_–A_2k−1_—the amplitudes of harmonics, and ω—the base frequency of a voltage signal.

To get a minimum of the criterion *S,* all partial derivatives dS/dA_i_ should be equalized to zero:(13)$$\left(\begin{array}{c} \begin{array}{c} \frac{\partial S}{\partial A_{1}}=\overset{k}{\underset{i=1}{\sum }}\sin\left({\omega t}_{i}\right)\left[V_{i}-A_{1}\sin\left({\omega t}_{i}\right)-\ldots -A_{2k-1}\sin\left(\left(2k-1\right){\omega t}_{i}\right)\right]=0;\\ \ldots \\ \frac{\partial S}{\partial A_{n}}=\overset{k}{\underset{i=1}{\sum }}\sin\left(\left(2n-1\right){\omega t}_{i}\right)\left[V_{i}-A_{1}\sin\left({\omega t}_{i}\right)-\ldots -A_{2k-1}\sin\left(\left(2k-1\right){\omega t}_{i}\right)\right]=0; \end{array}\\ \ldots .\\ \frac{\partial S}{\partial A_{k}}=\overset{k}{\underset{i=1}{\sum }}\sin\left(\left(2k-1\right){\omega t}_{i}\right)\left[V_{i}-A_{1}\sin\left({\omega t}_{i}\right)-\ldots -A_{2k-1}\sin\left(\left(2k-1\right){\omega t}_{i}\right)\right]=0; \end{array}\right)\;$$

The set (13) represents the system of k-linear algebraic equations, which can be solved in matrix form:(14)$$\left(\begin{array}{c} A_{1}\\ \begin{array}{c} A_{3}\\ \vdots \\ A_{2k-1} \end{array} \end{array}\right)={\left(\begin{array}{cc} \begin{array}{cc} a_{11} & a_{12}\\ a_{21} & a_{22} \end{array} & \begin{array}{cc} \cdots & a_{1k}\\ \cdots & a_{2k} \end{array}\\ \begin{array}{cc} \cdots & \cdots \\ a_{k1} & a_{k2} \end{array} & \begin{array}{cc} \cdots & \cdots \\ \cdots & a_{\mathrm{kk}} \end{array} \end{array}\right)}^{-1}\cdot \left(\begin{array}{c} U_{1}\\ \begin{array}{c} U_{2}\\ \vdots \\ U_{k} \end{array} \end{array}\right)$$
where:(15)$$U_{i}=\overset{N}{\underset{i=1}{\sum }}V_{i}\sin\left(\left(2n-1\right)\cdot {\omega t}_{i}\right)\;,\;n\in 1,\ldots ,k$$
(16)$$a_{\mathrm{lm}}=\overset{N}{\underset{i=1}{\sum }}\sin\left(\left(2l-1\right)\cdot {\omega t}_{i}\right)\;\sin\left(\left(2m-1\right)\cdot {\omega t}_{i}\right)\;,\;l,n\in 1,\ldots ,k$$

## 3. Results

Following the methods of the RMS voltage estimations presented here, the set of laboratory measurements was carried out to verify the accuracy of all submitted approaches. For the verification of the RMS measurement approaches and comparison between different suggested methods, the laboratory setup was created (Figure 3 and Figure 4). It includes the laboratory AC voltage source DRTS 33 [36], which is able to simulate real AC voltage with an arbitrary harmonic content, and the voltage transducer CV 3-500 [37], which translates high instantaneous network voltage amplitudes to low-voltage signal. The analog converter developed by authors scales and shifts the output voltage of the transducer CV 3-500 to the analog A/D input through the use of a Teensy 4.1 controller. The resistive components in the analog converter have the following magnitudes: R1 = 15.54 kΩ, R2 = 15.32 kΩ, R3 = 1 kΩ, and Rf = 9.88 kΩ.

The controller translates the analog signal to digital information and estimates the voltage RMS each 6–7 msec comparing it to the nominal value. The controller decides which reactive power value should be introduced to the load when the voltage level overcomes permissible boundaries. Moreover, the controller calculates a required capacitance, determines that the capacitors in the bank are suitable to be connected to a load, and sends the signals to the appropriate SCR drivers that are responsible for connecting/disconnecting specific capacitors. In addition, the laboratory equipment for supervising the experimental process and accepting decisions used a digital A/D 10-bit transducer digitizing 2000 uniformly distributed magnitudes per one period of the AC signal and acquiring data in the built-in memory.

The AC source simulated real voltage signal with a content from the first up to the twenty-fifth odd harmonic.

The magnitude of each harmonic was determined by the standard [38]. These are represented in Table 2.

### 3.1. Results of the RMS Estimation Due to the Amplitude of the Signal

This method, despite being simple and easily applicable, provides the lowest precision of the RMS assessment that is not influenced by the sampling time. This method’s error is a result of an instability of a voltage signal as well by a lack in the synchronization of the measuring amplitude magnitude. Altogether, for the existing A/D converters, the error value lies between 1.30–1.50%.

### 3.2. Results of the RMS Estimation by the Sampling of a Signal

The voltage magnitude sampling with correcting coefficient provides much more precision for the RMS assessment. However, this depends on the sampling time. Figure 5a–c shows a relative error histogram for the sampling times of 5, 7, and 9 ms.

The root mean square error with its standard deviation (STD) for this method is represented in Figure 6.

The error and its STD tend to decrease with a sampling time. The error for the suitable sampling time (6–7 ms) can be no more than 0.4–0.6%.

### 3.3. Results of the RMS Estimation by Decomposing the Signal to Odd Harmonics

The decomposition of a signal to odd harmonics theoretically should ensure the most accurate results of the RMS value. This approach, despite being accurate, requires significant computational effort that increases as the number of harmonics increases. Considering this circumstance, the RMS estimation error should be investigated in detail.

Histograms of relative error for one-harmonics RMS estimation with the sampling times of 5 ms, 7 ms, and 9 ms are represented in Figure 7. The same histogram form is for two- and three-harmonics signal decomposition.

The average relative RMS error for one-, two-, and three-harmonics representations are shown in Figure 8.

As in the previous method, the error decreases with the increase in sampling time and with more significant harmonics signal decomposition. The STD of an error can be even more than its average value and comprises 0.6–1%. This circumstance can be explained by the relatively low value of an average error having scattered statistics.

The accuracy of this approach for a one-harmonics representation is close to that of a previous method based on sampling voltage magnitudes. However, it became significantly more accurate with the use of two- or three-harmonics. The average error in this case for the (6–7 ms) sampling time is less than 0.25–0.35% substantially better than for the method based on voltage sampling in a fraction of the AC period time. 

## 4. Discussion

The requirement to keep voltage inside narrow boundaries over time has grown into an actual obligation during the last ten years. The need for fast voltage correction in distribution lines and other important locations in electric grids has become more and more tangible. The widespread fast-changing loads and stochastically generating plants produce a significant demand for finding appropriate and efficient solutions for preventing voltage deviations. The inevitable part of any control system aiming to keep a constant voltage level is the entity able to estimate the voltage RMS during a short time. The existing methods and means as a rule provide such a measurement during relatively long times, including many AC periods. The authors were motivated to find and analyze methods of RMS assessment in a time shorter than half the AC cycle duration. The main reason for such a task is the technical request of some sensitive consumers to ensure voltage constancy in each AC cycle.

The three most affordable approaches were studied in the proposed work. The first method is based on the RMS relation with the amplitude of a sinusoidal signal. Two other solutions were developed by the authors. The second method uses a restricted sampling time of voltage magnitudes in the fraction of the AC period with the application of a known RMS expression. However, this should be corrected by a special coefficient suggested in the work. The third approach is based on the decomposition of the voltage signal with the first n-odd harmonics. The well-known method of Fourier series representation can be applied. However, it is relatively heavy and less accurate compared with that developed in this work. The novelty of the submitted approach is in the description of a periodic (sinusoidal) signal by a set of harmonics that fits the obtained voltage points by the least-mean-square algorithm. This is based on solving the matrix equations. Such an approach prevents the need of a finding Fourier coefficient by integral calculations and in this way eliminates the requirements for applying strong computational efforts.

This investigation explored the accuracy of all analyzed approaches. The best accuracy was found to be in the novel method based on a signal decomposition. The error of the RMS estimation can be diminished up to 0.2–0.3%. However, this method requires significant computational power, which can complicate the development of a control system. A more convenient method can be developed by sampling voltage magnitudes during a fraction of the AC cycle (from 1/4 to 1/2 of a period) and applying a correcting coefficient depending on the sampling time. Its accuracy can be characterized by the relative error of 0.4–0.6%. Such accuracy is quite acceptable in most industrial applications. Altogether, it is worth emphasizing the simplicity of the equipment used to estimate the voltage RMS.

## 5. Conclusions

The present work is dedicated to finding a suitable solution for the AC voltage RMS assessment in a short time and with appropriate accuracy. The authors analyzed three affordable methods, two of which they developed. The most applicable approach is based on sampling voltage magnitudes during a fraction of the AC period and an applying a correcting coefficient depending on the time of the voltage measurement. This method can ensure a short time of the RMS estimation (less than half of the AC cycle) with an accuracy of 0.4–0.6%. Such estimation quality is acceptable enough for most modern electrical and electronic facilities.

## Figures and Tables

**Figure 1 sensors-22-06892-f001:**
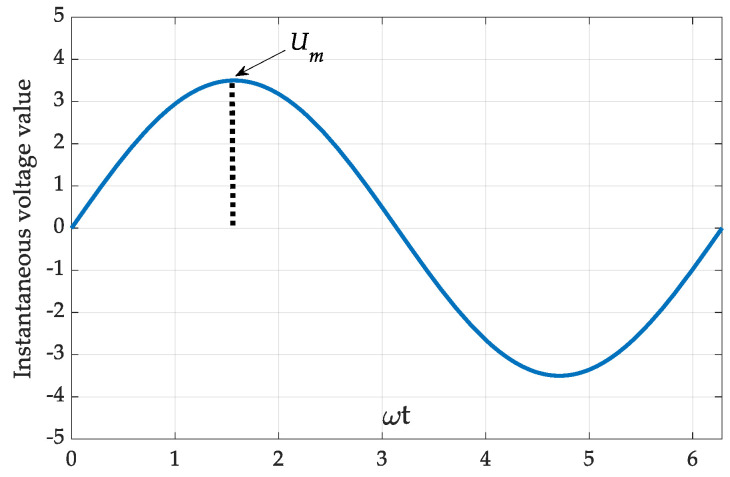
Instantaneous AC voltage curve.

**Figure 2 sensors-22-06892-f002:**
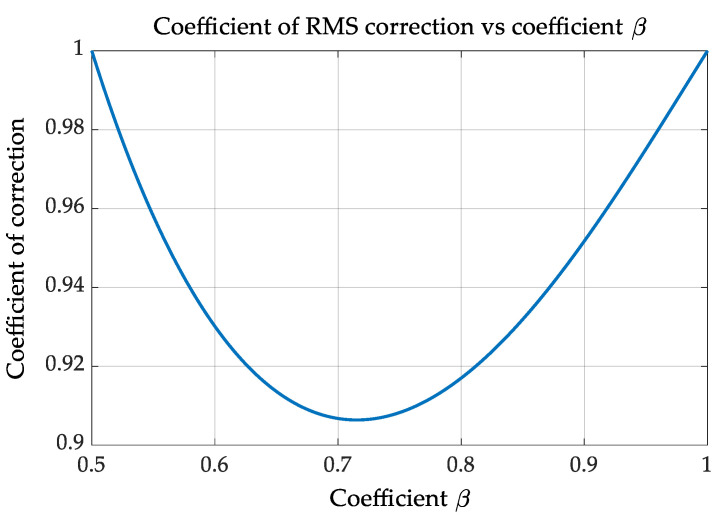
Coefficient of correction RMS estimation through the fraction of AC period.

**Figure 3 sensors-22-06892-f003:**
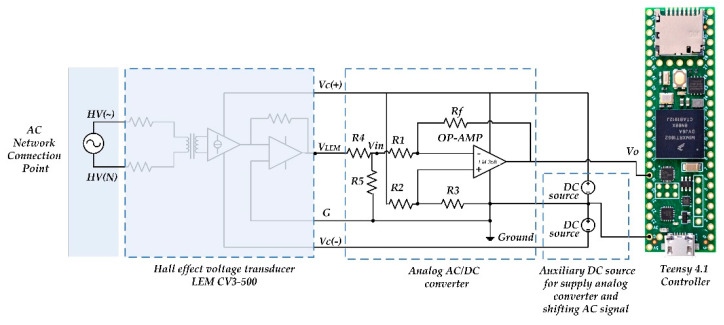
Principle scheme of the voltage transducer. For the first step from high to low voltage used the device LEM CV-3-300 [37].

**Figure 4 sensors-22-06892-f004:**
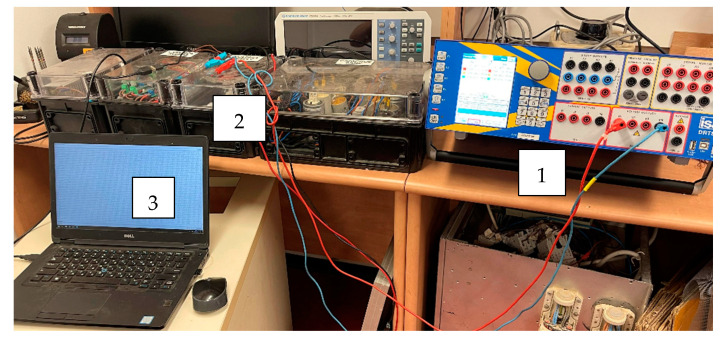
Test-bench for the verification different methods of the RMS estimation: 1—AC voltage source; 2—voltage transducer; 3—computer for data recording.

**Figure 5 sensors-22-06892-f005:**
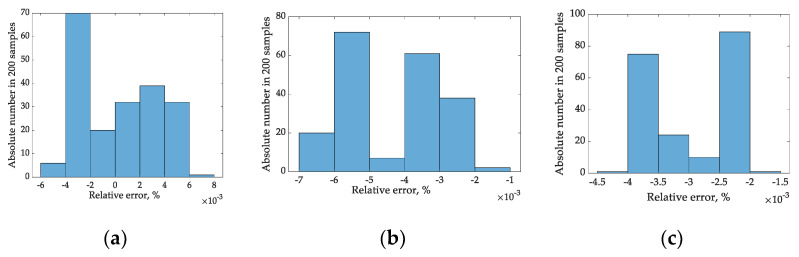
Histograms of a relative RMS estimation error for sampling time 5 ms (**a**), 7 ms (**b**), and 9 ms (**c**).

**Figure 6 sensors-22-06892-f006:**
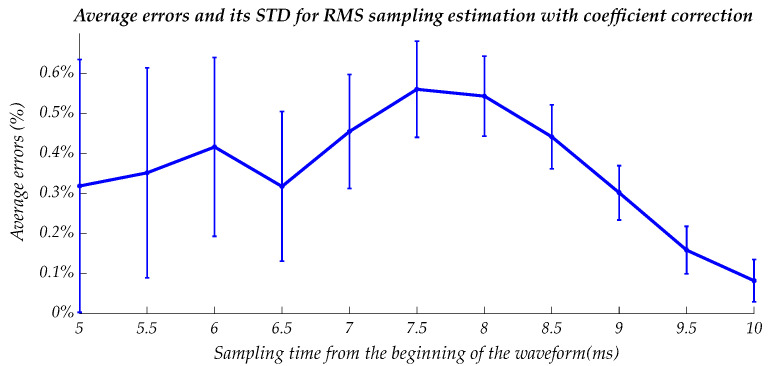
Estimation error of the method based on a sampling instantaneous voltage magnitude.

**Figure 7 sensors-22-06892-f007:**
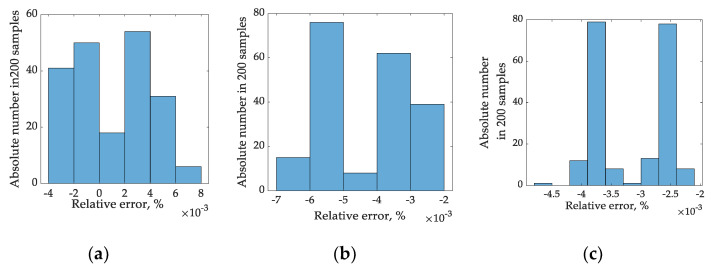
Histograms of a relative RMS error for one-harmonic estimation with sampling time of 5 ms (**a**), 7 ms (**b**), and 9 ms (**c**).

**Figure 8 sensors-22-06892-f008:**
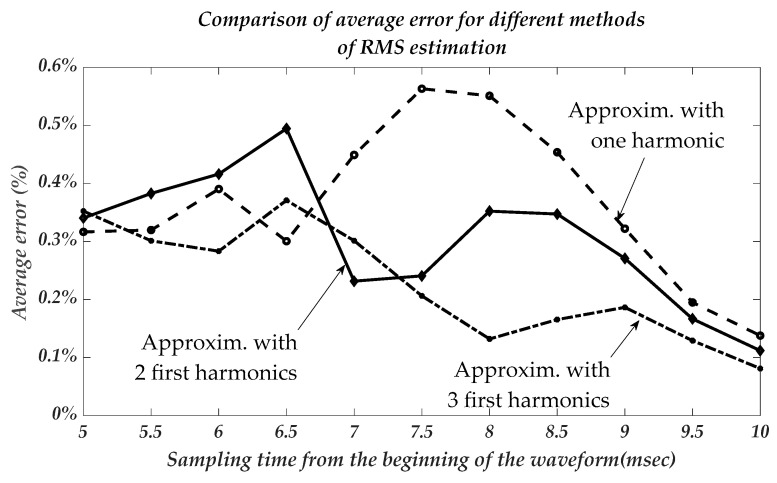
The average relative route-mean-square error for one-two, and three-harmonics representations as per sampling time.

**Table 1 sensors-22-06892-t001:** Comparing between different methods of a fast RMS estimations.

N	Method	Complexity of Implementation	Author/Reference	Precision	Estimation Time Relative to a Time of AC Period (T)
1.	Conventional RMS estimation by instantaneous voltage magnitude during integer numbers of AC periods	Middle	Thakur and Singh [31]	High	Long (5 ÷ 7) T
2.	DC component measurement of rectified AC signal	Low	Muciek [26]	High	Very long (10 ÷ 15) T
3.	Analysis of the main signal harmonic	Middle	Naidoo and Pillay [32]	Average	Long (6 ÷ 8) T
4.	RMS estimation through signal amplitude	Very low	Naidoo and Pillay [32]	Very low	Very short (T/4 ÷ T/2)
5.	Based on instantaneous voltage magnitudes during fraction of AC period with correction coefficient	Middle	Developed in this article	High	Very short (T/4 ÷ T/2)
6.	Decomposition of voltage signal by the ensemble of harmonics	High	Developed in this article	High	Very short (T/4 ÷ T/2)

**Table 2 sensors-22-06892-t002:** Relative magnitudes of harmonics in a signal modeling real voltage signal.

% of the Full Voltage												
**U_1_**	**U_3_**	**U_5_**	**U_7_**	**U_9_**	**U_11_**	**U_13_**	**U_15_**	**U_17_**	**U_19_**	**U_21_**	**U_23_**	**U_25_**
99.4	5	6	5	1.5	3.5	3	0.5	2	1.5	0.5	1.5	1.5

## Data Availability

Not applicable.

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
