# Peer review of "Voltage RMS Estimation during a Fraction of the AC Period"

_sensors, 2022, doi:10.3390/s22186892_

Round 1
Reviewer 1 Report
The authors propose an approach for RMS estimation during a fraction of the AC period. The paper is well written and easy to follow. The test and results are sound and well analyzed, the conclusions are based on what is presented in the results. In general terms the paper is well organized; nonetheless, there are some minor details that the authors must consider to improve the presentation of the document:
1- Towards the end of the introduction the authors may provide a table summarizing the main contributions/features of other related research work mentioned in the bibliographical review. This might enable the reader to easily identify the research gap of the proposed work.
2- All tables must have a sentence that explains what they present (not just Table X. )
3- In figures 4, 6 and 7 it is difficult to read the information of the axes (please change the size of the font).
4- Please provide a brief summary of the content of the paper at the end of the introduction.
5- Please list the main contributions of the document towards the end of the instruction. This implies rewriting lines 100-108
Reviewer 2 Report
This paper presents and proposes some methods to estimate the RMS value of a grid AC voltage. When there is an imbalance between energy production and its consumption, the grid voltage gets altered. The authors claim that the proposed methods can make an estimation about the RMS voltage in less than half of a period, so that the observed deviation can be promptly tackled making a proportional use of reactive elements. A literature review was undertaken, to introduce the subject and to locate it in its field of knowledge. In addition, an experimental setup was established, which greatly improves the manuscript content, given that the results were obtained from laboratorial experiments. The overall results show encouraging values. The quality of the written English is good, although needing some local corrections.
However, there are some remarks regarding the content of the paper. The main issues are concerned about correcting some typo mistakes that, in a paper with the purpose of having worldwide visibility, should not occur. Also, there is the need for clarification of some aspects, as follows:
Mark 24 – “become” should be replaced with “became”.
Mark 49 – “…[10]. Whereas…” should be replaced with “…[10], whereas…”.
Mark 56 – “…control. Whereas…” should be replaced with “…control, whereas…”.
Mark 120 – “his” should be replaced with “this”.
Mark 124 – “t.i.” should be replaced with “i.e.”.
Mark 130 – “Assuming” should be replaced with “Assume”.
From this point on, the authors need to carefully check the numbering of the equations, since there are inconsistencies, both in its order and when referred to in the main text.
Mark 144 – “+–” should be replaced with “+/–”.
Mark 145 – “intervene” should be replaced with “inject”.
Mark 147 – “a” should be inserted before “dimensionless”. According to the text, the equation that follows should be using beta, instead of theta.
Mark 150 – “it” should be inserted before “must”.
Mark 152 – “tmes” should be italicized, to match with the symbol in the equation that follows and “??? ?? ????????? 50??” shouldn’t be italicized.
Mark 153 – “Figure 1” should be “Figure 2”. The numbering of the figures from this point on should be corrected.
Mark 157 – “pointduring” should be replaced with “points during”.
Mark 164 – “is” should be replaced with “it will be”.
Mark 165 – “to” should be inserted before “calculate”.
In the first figure of section 3, the operational amplifiers in the schematic should have their input terminals (inverting and non-inverting) properly identified. Moreover, the schematic should be clearly explained, labeling all the components, and showing their value. The role of the Teensy controller should also be clearly explained (program being run, pins being used, etc.).
In the results that follow, one should have an expectation of their value, concerning the desired RMS value. It would be helpful to say something about the amplitude of the AC voltage under analysis, and the expected RMS estimation by each of the methods.
Mark 193 – The % symbol is duplicated. How were the presented values obtained? What computations led to those values? This should be clearly shown.
Mark 205 – “…decomposition signal…” should be replaced with “…decomposing the signal…”.
Mark 219 – “that” should be replaced with “what”.
Mark 227 – “become” should be replaced with “became”.
Mark 238 – “…one of…” should be replaced with “…of the…”.
Mark 244 – “compare” should be replaced with “compared”.
Mark 246 – “which” should be removed.
Mark 248 – “…a finding…” should be replaced with “…finding a…”.
Mark 254 – “voltages” should be replaced with “voltage”.
Mark 265 – “applying” should be replaced with “applied”.
Round 2
Reviewer 2 Report
This paper presents and proposes some methods to estimate the RMS value of a grid AC voltage. When there is an imbalance between energy production and its consumption, the grid voltage gets altered. The authors claim that the proposed methods can make an estimation about the RMS voltage in less than half of a period, so that the observed deviation can be promptly tackled making a proportional use of reactive elements. A literature review was undertaken, to introduce the subject and to locate it in its field of knowledge. In addition, an experimental setup was established, which greatly improves the manuscript content, given that the results were obtained from laboratorial experiments. The overall results show encouraging values. The quality of the written English is good.
After analyzing this re-submission, and as far as I’m concerned, the comments to the first submission have been mostly addressed. The additional content that has been elaborated helped the paper to get more clearer, robust, and enriched. However, there are still some typo details, among some others, like duplicate periods (.), in the content of the paper that should be addressed in this re-submission, as follows:
Mark 58 – Some attention is needed in the writing.
Mark 219 – “deposing” should be replaced with “decomposing”.
Mark 231 – “sighificant” should be replaced with “significant”.
Mark 272 – “Alogether” should be replaced with “Altogether”.
As such, I foresee that the actions to be taken by the authors should be to comply with the remarks that still subsist. From my point of view, the paper can be published when this is fixed and provided that the conditions imposed by the remaining reviewers are fully met.
Author Response
Answers for Reviewer N2 after the second revision.
Thank you, once again, for your excellent work.
Here are placed your suggestions with our answers.
Mark 58 – Some attention is needed in the writing.
Thank you. O.K. I will consider this suggestion.
Mark 219 – “deposing” should be replaced with “decomposing”.
Thank you. It was done.
Mark 231 – “sighificant” should be replaced with “significant”.
Thank you. It was done.
Mark 272 – “Alogether” should be replaced with “Altogether”.
Thank you. It was done.
Thank you once again!
